# Effects of Menstrual Health and Hygiene on School Absenteeism and Drop-Out among Adolescent Girls in Rural Gambia

**DOI:** 10.3390/ijerph19063337

**Published:** 2022-03-11

**Authors:** Vishna Shah, Helen Nabwera, Bakary Sonko, Fatou Bajo, Fatou Faal, Mariama Saidykhan, Yamoundaw Jallow, Omar Keita, Wolf-Peter Schmidt, Belen Torondel

**Affiliations:** 1Environmental Health Group, Department of Infectious Diseases, London School of Hygiene and Tropical Medicine, London WC1E 7HT, UK; wolf-peter.schmidt@lshtm.ac.uk (W.-P.S.); belen.torondel@lshtm.ac.uk (B.T.); 2Department of Education and Clinical Sciences, Liverpool School of Tropical Medicine, Pembroke Place, Liverpool L3 5QA, UK; helen.nabwera@lstmed.ac.uk; 3The Medical Research Council Unit the Gambia, London School of Hygiene & Tropical Medicine, Banjul P.O. Box 273, The Gambia; bsonko@mrc.gm (B.S.); fbajo7829@gmail.com (F.B.); ffaal@mrc.gm (F.F.); masaidykhan@mrc.gm (M.S.); yajsamba@mrc.gm (Y.J.); 4Regional Education Directorate Four, Ministry of Basic and Secondary Education, Mansakonko Lower River Region, Banjul P.O. Box 989, The Gambia; keitaomar1984@gmail.com

**Keywords:** menstrual hygiene management, adolescence, menstrual knowledge, young people, school absenteeism, school attendance

## Abstract

Poor knowledge and management of menstruation impacts girls’ school attendance and academic performance. This paper aims to explore how menstrual hygiene management practices and related factors influence school absenteeism and drop-out among primary and secondary school girls in rural Gambia. Mixed-method studies were conducted among students and key informants from 19 schools from July 2015–December 2017. Focus group discussions, in-depth interviews, cross-sectional surveys, menstrual diaries, and school water, sanitation and hygiene (WASH) facility observations were used. Key findings from the interviews were that menstrual pain, cultural beliefs, fear of peers knowing menstrual status, and poor school WASH facilities led to school absenteeism, however, they had no impact on school drop-out. Of the 561 girls surveyed, 27% reported missing at least one school day per month due to menses. Missing school during the most recent menstrual period was strongly associated with menstrual pain (extreme pain adjusted odds ratio (AOR) = 16.8 (95% CI: 7.29–38.74)), as was having at least one symptom suggestive of urinary tract infection (AOR = 1.71 (95% CI: 1.16–2.52)) or reproductive tract infection (AOR = 1.99 (95% CI: 1.34–2.94)). Clean toilets (AOR = 0.44 (95% CI: 0.26–75)), being happy using school latrines while menstruating (AOR = 0.59 (95% CI: 0.37–0.93)), and soap availability (AOR = 0.46 (95% CI: 0.3–0.73)) were associated with reduced odds of school absenteeism. This study suggests menstrual pain, school WASH facilities, urogenital infections, and cultural beliefs affected school attendance among menstruating girls in rural Gambia.

## 1. Introduction

Several studies across low and middle income countries (LMICS) suggest more than 50% of girls have inadequate menstrual hygiene management (MHM) practices, with higher proportions in rural areas [1,2,3,4]. Poor MHM may lead to school absenteeism, disengagement or withdrawal from school [1,5], reproductive problems, urinary tract infections [5,6,7], withdrawal from social gatherings, and unnecessary restrictions to the daily routine [8].

Studies show a wide range in reported levels of school absence due to menstruation. A survey in Indonesia reported 11% of adolescent girls missed school during their last period [9], in Niger, Burkina Faso, and Nigeria, 15%, 17%, and 23% of girls aged 15–24 years reported missing school in the past year due to menstruation [10]; an earlier survey in Nigeria found 43% of female university students reported missing school due to menstrual pain [11]. In Ethiopia, 43–50.7% of students reported being absent from school due to menstrual-related problems [12].

Regular school absenteeism can affect school performance, self-esteem, and ability to pursue higher levels of education [1,5]. Withdrawal from school may result in early marriage, teenage pregnancies, increased risk of sexually transmitted infections, and loss of opportunities [13,14,15,16]. Studies globally, have shown an extra year of secondary school can increase a girl’s potential income by 15–25%, and a 1% increase in a girl’s secondary school attendance can boost a country’s GDP by 0.3% [17]. Better educated women are more likely to be healthier and have better health status for their children than uneducated women, they also tend to have formal employment, earn higher wages, marry later, have fewer children, and prioritise their children’s education. This can reduce poverty and contribute to community and country development [18,19,20,21].

There are a range of barriers to school attendance and engagement that have been identified in the literature, and can broadly be split into two categories: physical aspects and psychosocial aspects. Physical aspects include poor access to school water sanitation and hygiene (WASH) facilities, lack of privacy, poor access to clean and effective absorbent materials, lack of absorbent disposal facilities, and lack of pain killers to mitigate menstrual pain [9,10,11,12,22,23,24,25]. The psychosocial aspects stem from the cultural taboos and stigmas associated with menstruation, which hindered adolescents from seeking advice from parents and teachers on appropriate MHM practices [26], resulting in poor knowledge about menstruation and its management and poor social support [10,27,28,29,30,31], which results in episodes of teasing, stress, embarrassment, and lack of confidence when menstruating in school [10,27,28,29,30,31].

Systematic reviews have shown a lack of evidence for the effectiveness of menstrual management interventions among schoolgirls on reducing school absenteeism and drop-out [1,5]. The “MHM in Ten” group represented by United Nations agencies, non-governmental organisations, and academics and stakeholders was set up in 2014 to set out priorities for improving menstrual hygiene management in schools. They recognised there is an urgent need to generate robust evidence on the impact of inadequate MHM on health and educational outcomes among girls [8,32].

In The Gambia, there is limited evidence on how menstruation affects school attendance among girls. Our prior work in rural Gambia found that taboos, secrecy, and shame were prevalent when discussing menstruation. The study showed that students (boys and girls) had poor knowledge about menstruation, largely due to the need for secrecy and cultural beliefs [26]. The Gambian government recognises the need to invest in initiatives to improve MHM among school-going girls through the Ministry of Basic and Secondary Education (MoBSE), one initiative of theirs is to provide free menstrual pads in all government schools to students in grade six and above [33]. However, more research is needed to better understand the levels of absenteeism due to menstruation and factors associated to it. This information will provide better understanding on how new interventions can be designed to tackle this problem.

The objectives of this study are: (i) to explore knowledge, attitudes, and practices related to menstruation, (ii) to describe the school’s WASH facilities, (iii) to measure levels of school absenteeism due to menstruation, and (iv) to explore how factors linked to menstruation are associated with school absenteeism and drop-out among rural Gambian schoolgirls.

## 2. Materials and Methods

### 2.1. Study Setting and Population

This study was conducted in 19 rural primary and secondary schools located in 13 villages in the rural Kiang districts, in the Lower River Region of The Gambia (Bajana, Jali, Janneh Kunda, Jiffarong, Kaiaf, KantongKunda, Karantaba, Keneba, Kwinella, Manduar, Nema Kuta, Niorro Jattaba, and Sibito) between July 2015 and December 2017. Schools were selected through recommendation of the MoBSE. Of the 19 schools, twelve were English schools and seven were Arabic schools. The Gambia has two formal education systems: Arabic-based and English-based. The English-based schools are free public schools, and the Arabic schools are private schools that focus on Quranic education. Schools in The Gambia also run a double shifting programme, where some grades attend school in the mornings, while other grades attend schools in the afternoon. This is partly due to teacher and resource constraints, but also to allow students to carry out domestic duties [34].

The population in the Kiang district is predominantly from the Mandinka ethnic group, the majority are Muslims and most families are polygamous [35]. The majority of the population live below the moderate poverty line of less than USD 2/day [36,37].

### 2.2. Informed Consent

Before enrolment, informed consent was obtained from students over 18. Assent was sought from younger students and consent was given by their carers. It was made clear to all participants that their participation was voluntary and would not affect any other services they receive [38].

### 2.3. Study Design and Sampling

Data presented in this paper are part of mixed method studies that involved two cross-sectional surveys, qualitative interviews (focus group discussions (FGDs) and in-depth interviews (IDIs)), daily diaries, and unannounced WASH spot checks. A mixed methods approach was used so as to benefit from both the detailed contextualized insight from the qualitative data and the generalizable externally valid insights from the quantitative data. The study started with the qualitative interviews to be able to set the scene and have a better understanding of the MHM context in The Gambia, as not much was known about it; qualitative methods were also seen to be useful to explore some of the aspects related to menstruation due to the taboo nature of the topic. The qualitative interview guides were developed through systematic reviews on MHM and through discussions with the local study team and local clinicians to explore what challenges needed to be addressed. The points raised in the interviews and existing tools used to evaluate MHM outcomes were used to develop the cross-sectional survey tools. Appendix A show the qualitative interview guides and cross-sectional survey tools. The qualitative results were also used to triangulate findings with the quantitative data and explore issues that are not possible to address through a survey. Triangulation of methods can provide opportunities for testing alternative interpretations of data and for examining the extent to which the context helped to shape the quantitative results.

#### 2.3.1. Qualitative Interviews Sampling

A subset of six schools were selected to conduct the qualitative interviews in. Selected schools were a representative of the other schools in the region. We aimed to conduct at least one FGD with each of the following groups, pre-menarche girls, post-menarche girls, and boys in each of the six schools. For the IDIs, we aimed to have at least two IDIs among each of the following categories: girls who had dropped out of school, boys, mothers, and teachers, across all six schools. Additional discussions were conducted if new themes continued to immerge through the discussions. Participants for the FGDs and IDIs were purposively selected. A group size of between 5–8 participants was considered adequate for the FGDs, discussions with fewer than five people were not conducted to ensure the anonymity of participants was not risked. The school teachers were asked to provide a list of different groups listed below that they considered able to discuss sensitive topics and were engaged during discussions: girls between 13–20 years, boys between 15–20 years, mothers who had adolescent children in the school, and drop-out female students whose age at drop-out was 15 years or older. Participants were randomly chosen from these lists. Teachers who were interested in the study topic and worked closely with students on topics related to reproductive health were invited to participate in the IDIs. Participants who were considered open, and had a lot of information to share that could not be captured during the FGDs, were invited to continue the discussion during an IDI.

#### 2.3.2. Cross-Sectional Surveys Sampling

All participating schools (*n* = 19) were asked to create lists of female students that were 13 years and older from grades 5–12, all girls from this list were asked their menstrual status, and those that had started menstruating were included in the survey. Sample size calculations for the individual cross-sectional studies are described elsewhere [6,26].

#### 2.3.3. Daily Diaries Sampling

In order to compare different methods of measuring school attendance related to menstrual status, daily diaries were given to a subset of participants. Participants for the daily diaries were randomly selected. Study IDs of all girls consenting from four schools (2 English and 2 Arabic) were split by age into three categories: 11–14 years, 15–17 years, and 18–25 years. IDs from each category were entered into a separate random number selector, 10 numbers were selected from each category to have a total of 30 numbers selected. The categories were created to ensure there was representation from each age category.

#### 2.3.4. Unannounced WASH Spot Checks

All schools enrolled in the study (*n* = 19) received unannounced WASH spot checks.

### 2.4. Data Collection

#### 2.4.1. Qualitative Interview Data Collection

Interviews were conducted in the local language, using semi-structured interview guides, which contained open-ended questions and suggested probing questions. Interviewers were of the same gender as participants to ensure comfort and openness of participants. Girls were asked their menarcheal status on a one-to-one basis prior to the FGDs, and then split into groups of pre- and post-menarche girls. FGDs consisted of participants, a moderator, and a note taker; the moderator led the discussions, the note taker was there to note any non-verbal cues, highlight important topics that came up in the discussion, and assist the moderator. IDIs included the participant and moderator.

The discussions explored knowledge and attitudes towards menstruation, cultural norms associated with menstruation, MHM practices at home and school, challenges faced managing menses, opinions on school WASH facilities, reasons for school absenteeism when menstruating or dropping out of school, role of parents and teachers on menstrual support for adolescents, and potential solutions to supporting girls while they menstruate.

FGDs and IDIs were conducted in a private room and recorded using a digital voice recorder. The IDI lasted about 25–40 min and the FGDs lasted about 30–70 min. Due to the sensitive nature of the topic, moderators spent some time at the start of the discussions creating a rapport with the participants. Data saturation was the marker of adequate sampling [39].

#### 2.4.2. Cross-Sectional Survey Data Collection

Enumerator-administered pre-tested questionnaires were used due to low literacy levels and the local language (Mandinka) not being a written language. Data on socio-demographics, knowledge and practices of menstruation (including age of menarche; knowledge and attitudes about menstruation; MHM practices such as type of menstrual absorbent used, change frequency, washing and drying practices of reusable material; access to WASH at home), school absenteeism, reasons for absenteeism, and symptoms of urogenital infections were collected. Questions about school absenteeism referred to the last 30 days, and the questions were subdivided to ask about all frequently reported reasons for absenteeism. To assess school attendance related to menstruation, the question “in the last 30 days, how many days did you miss of school because of your period?” was used. Data completeness and consistency were reviewed at the end of each data collection day by the team’s supervisor (VS).

#### 2.4.3. Daily Diaries Data Collection

The daily diaries given to the subset of 30 random girls were to be completed daily by the girls over an 8-week period, indicating if they attended school or not and if they were menstruating or not that day (Appendix A—soft copy of diary distributed). The enumerators indicated the girls study ID on the diary and explained how to fill in the diaries and completed one week retrospectively to give the girls an example of how to complete it. A month later, the team then went again to check if the girls needed any help and then again at the end to collect the diaries.

#### 2.4.4. WASH Spot Checks Data Collection

Unannounced visits to conduct spot checks of WASH facilities were conducted in each school by the team’s supervisor. A pre-determined tool was used to assess the WASH hardware available: number and type of toilets, handwashing facilities, and disposal facilities. As well as access, the quality of these facilities, functionality, gender specificity, privacy, cleanliness, availability of water and soap, and location of water was assessed. Permissions to conduct unannounced visits were obtained from each school at the beginning of the study. One spot check was conducted in each school over the course of the study period. Spot checks were conducted during school hours to ensure the facilities could be observed in conditions that would be available for students. The study team also ensured there was a lag between obtaining consent for unannounced visits and the actual visit.

#### 2.4.5. Pilot Testing of Tools

Questions used in tools (interview guides and survey questionnaires) were put together from several different sources that had been validated, however, since the questions had not been tested/validated in the combination we used and in the Gambian context, we piloted the tools we developed for the study on 11 volunteers (3 medical practitioners, 4 field assistants from MRCG Keneba, 3 adolescent girls, and 1 adolescent boy). The tools were first piloted on the medical practitioners and field assistants and feedback from these groups was used to amend the tools; after these groups were satisfied with the tools, they were then piloted on the adolescents. The tools had final adjustments after feedback from the adolescents and were then considered valid for this context. The main aim of the piloting was testing the quality and acceptability of questions and their translation, and the feasibility of using the tools. Other benefits from the piloting were they supported enumerator training in using and administering the tools.

The WASH spot check tool was adapted from the UNICEF international guidelines [40]; these guidelines set out questions and indicators to standardize monitoring WASH in schools in line with the sustainable development goals. The adapted tool was piloted to test if it worked well in the Gambian context.

### 2.5. Data Management and Analysis

#### 2.5.1. Qualitative Data

Data from the IDIs and FGDs were simultaneously translated and transcribed into English by the field team. Inductive content analysis was conducted [41]. VS analysed all the transcripts. Six randomly selected transcripts were assigned to the HN for analysis to test the inter-rater reliability regarding the codes and themes emerging from the transcripts. HN and VS independently read the transcribed data carefully and segmented the data. Each researcher assigned meaningful segments a code and the codes were then discussed and compared.

#### 2.5.2. Survey Data

Data collected on questionnaire forms were double entered into SQL (SQL server 2017 [42]) and analysed using Stata version 16.0, [43]. The cross-sectional data was presented by school type and comparisons made using chi-squared statistics for binary and categorical outcomes and t-test for continuous outcomes. Principal component analysis (PCA) was used to determine household socio-economic status using an asset-based index [44]. The adolescent girls’ households were classified into three quantiles (i.e., poorest/2nd quantile/least poor) based on Filmer and Pritchett’s method [45].

Knowledge of menstruation was assessed using five questions. Each question was scored on a scale of 0 (incorrect) to 1 (correct). The score for each respondent was totalled, with a maximum score of 5. A score of 0–1 was coded as poor knowledge and 2–5 was coded as good knowledge.

Univariable and multivariable logistic regression analyses of outcome variables were applied to provide both unadjusted and adjusted odds ratios in exploring factors associated with school absenteeism due to menstruation. In the multivariable regression analysis, a model was run for each of the MHM health outcomes and WASH variables. The variables age, wealth, school type, and maternal education were included as a priori confounders in the models, as they were associated with the outcome. School level data were adjusted for clustering using robust standard errors. Missing data were not imputed.

#### 2.5.3. Daily Diary Data

The study ID was used to link survey and diary data. From the diary, total number of days missed, and number of days missed due to menstruation in the 30-day prior to survey date were extracted and compared with survey data.

#### 2.5.4. WASH Spot Check Data

Data was collected on forms and double entered into SQL [42] and analysed using Stata 16 [43]. Basic descriptive analysis was carried out to describe the quality of WASH facilities per school type, after which univariable analysis was carried out to see if the quality of WASH components affected attendance.

## 3. Results

Results from the qualitative and quantitative aspects complemented each other and are presented in three main themes: (1) menstrual knowledge, practices, attitudes, and health outcomes; (2) school WASH facilities; and (3) school attendance. There is a further theme regarding reasons for dropping out of school which was captured only through qualitative methods.

### 3.1. Characteristics of Study Participants

Twenty FGDs and twenty-one IDIs were conducted between July–December 2015, sampling a total of 155 participants (Table 1). Of the 610 girls that were considered eligible for the study (i.e., had started menstruating), 561 (92%) participated in the survey (Figure 1) between November 2015 and June 2017. Table 2 shows the sociodemographic characteristics of the girls surveyed.

All participants were Muslim. A greater proportion of mothers had no formal education (31%) vs. fathers (16%). About two-thirds of the girls reported that their households’ main source of income was farming (67%). Only 4% of the girls reported having access to a household water source, the majority accessed water through communal standpipes (69%), and time to collect water was less than 15 min (83%). Most girls had access to an improved toilet facility (flush/pour flush toilets or pit latrines with a slab) at home (87%) (Table 2).

### 3.2. Menstrual Knowledge, Practices, Attitudes, and Health Outcomes

#### 3.2.1. Qualitative Findings

All groups interviewed had some knowledge about menstruation, however, levels of knowledge differed among the participants, surprisingly, the most accurate answer for what menstruation is came from a boy. The students showed many misconceptions and gaps in knowledge. This could largely be linked to the strong cultural norms of keeping menstruation private.


*“Special water comes out of the girl”*

*(Boys-FGD4)*



*“The blood comes from the middle of your head and then comes down to your private parts”.*

*(Pre-menarche girls-FGD3)*


The need for privacy and secrecy about discussing menstruation resulted in a lot of embarrassment and shame about the topic. Even during the discussions where only girls and women were present, girls were seen trying to hide their face or shying away when asked questions.


*“I didn’t tell anyone, I didn’t feel I could tell anyone, one of my friends found out so I told her, but I never told my parents.” *

*(Post-menarche girls FGD5)*



*“The moment you hear someone say period, even a woman, you feel ashamed, because it has to be something secret” *

*(Post-menarche girls-FGD3)*



*“They (boys and men) should not know…it is not important…It is not their way. It is a way for women.” *

*(IDI mother1)*


Other cultural beliefs such as “*isolating from others*” (Post-menarche girls-FGD4]) or “*If you are menstruating you should try avoid certain things, for example I try to limit my movement and stay home*” (IDI-dropout1) may have an impact on school attendance.

#### 3.2.2. Quantitative Findings

The mean age of menarche in this population is 14 years (SD = 1.67). Learning about menstruation prior to menarche was reported by 61% of the girls (Table 3). About two-thirds (65%) of the girls had good knowledge of menstruation. Knowing that pregnant women (83%) and old women (72%) do not menstruate were among the best answered knowledge-based questions. By contrast, only about half of the girls knew menstruation was not a disease (55%) or that menstrual blood came from the uterus (49%). Menstrual knowledge was seen to be better in girls attending English-based schools than Arabic-based schools (70% vs. 54%) (Table 3).

Almost half of the girls reported using a combination of reusable cloth and disposable pads (44%); reusable cloth was the next most frequently reported absorbent material used (35%), followed by disposable pads (21%). However, 60% of girls went to a school that provided free disposable sanitary pads. The number of girls reporting they knew the school gave free pads/were able to collect free pads from school was lower in the English schools than expected (83% rather than 100%). The lower use of disposable pads could be due to absorbent preferences or the inability to obtain enough pads, as 26% reported being unable to buy them (Table 3).

Girls reported having good practices of handling reusable menstrual absorbents. Most washed their absorbents with water and soap (96%) and dried the material under the sun (82%). However, qualitative interviews suggested that girls tried to hide their material when drying. Over half the girls reported changing their menstrual absorbent three times a day or more on their heaviest bleeding day, with a higher percentage in English-based schools (60%) than Arabic-based schools (48%) (Table 3). Reporting at least one symptom indicative of UTI were reported by 39% of the participants, with 47% reporting at least one symptom indicative of RTI (Table 3).

### 3.3. School WASH Facilities

#### 3.3.1. Qualitative Findings

During qualitative interviews themes of school toilets being unclean, “smelly” and not used while menstruating, came across in most discussions. Girls were reluctant to use the school toilets and preferred to use the ones at home.


*“When I was in school, the toilet facilities were very poor compared to now.... In those days there was only one toilet for both girls and boys. At that time also, the toilet was not clean and it also smells badly…It made you not want to use it at all” *

*(IDI-drop-out3)*



*“If you start menstruating in school you ask for permission to go home to clean yourself” *

*(post-menarche girls-FGD4)*


#### 3.3.2. Quantitative Findings

All schools had sanitation and water facilities, however, the quality of facilities varied across schools (Table 4). Gender specific toilets were seen in 95% of the schools. Having at least one clean toilet cubicle for girls was seen in 89% of the schools. All English-based schools had at least some toilet cubicles that provided privacy, however, only 43% of the Arabic schools had a door on at least one toilet cubicle (*p* = 0.003) and only 14% had a lockable door (*p* < 0.001). Less than a quarter (21%) of the schools surveyed had water inside at least one toilet cubicle. All schools had some form of handwashing facilities, the most common being a hand-pour system (84%), while others had running water from a standpipe or tubewell (54%). Soap at handwashing facilities were less common, about a third of schools (37%) had soap at handwashing facilities at the time of visit. Overall, 62% of girls reported being unhappy to use the school latrines while menstruating, with a greater proportion of girls in Arabic schools reporting being unhappy (76% vs. 58%) (*p* = 0.003) (Table 4).

### 3.4. School Absenteeism

During both qualitative and quantitative interactions with girls, there was an indication that girls missed school while menstruating. Missing at least one day of school per month due to menstruating was reported by 27% of the girls surveyed (Table 5). Menstrual pain, changing absorbents, fear of staining and smelling, school WASH facilities (described above), and keeping away from others while menstruating were the main factors reported for school absenteeism.

#### 3.4.1. Qualitative Findings

During interviews with girls, menstrual pain was the most frequently reported cause for missing school or affecting performance or concentration in school.


*“Yes, because on those days I don’t even go to school, because it (pain) disturbs me” *

*(IDI drop-out2)*



*“When a girl is menstruating she can feel stomach pain, if that happens the girl will not be able to concentrate in class” *

*(Post-menarche girls-FGD5)*


Most girls reported they would go home to change their menstrual absorbents. In some cases, girls were told by their teachers to take their friends to accompany them. The discussions made it seem that this was the case because they were uncomfortable changing in the school toilets, or for those using the reusable pads, they were uncomfortable changing their material and putting the used material in their bag, whereas at home they could wash it straightaway.


*“The teacher will ask another girl to accompany her to home if she is menstruating.” *

*(Boys-FGD3)*



*“When I saw my period in school I was not prepared that day. I was not expecting it. I had some stomach pain, but did not think it was menses. I didn’t carry any material with me… so when I saw it, I went home and used the bathroom to clean myself” *

*(IDI-drop-out2)*



*“These cloth, you don’t wash them in school, you can only wash them at home in the bathroom…so I don’t change it in school, I come home to change it” *

*(Post-menarche girls-FGD3)*


However, there were mixed opinions on whether girls would go home and change their absorbent material if they were provided absorbents at school.


*“If at all you are in school and your menses starts, you have to go to that teacher and ask for a pad, then you get permission (from the classroom teacher) to go home and you go prepare yourself.” *

*(Post-menarche girls-FGD3)*



*“If you are in school and you happen to start menstruating and you are given a pad, you can go to the toilets in school and put it on, rather than going home.” *

*(Post-menarche girls-FGD4)*


Another common reason for missing school while menstruating was fear that others would know they were menstruating, either because they may stain their clothes, or they may have bad odour or because they are not praying.


*“Some girls are ashamed to go to school while menstruating, because they think the blood will come out and people will know she is menstruating” *

*(Post-menarche girls-FGD4)*



*“Girls get teased sometimes, especially when it comes to prayer time and they refuse to pray, it makes girls very ashamed so many don’t come to school” *

*(IDI-teacher1)*


Girls mentioned several changes in the school environment that could improve their menstrual experience in school and reduce school absenteeism, namely access to pain relief strategies, access to water and soap in the toilets, and better access to absorbent material.


*“There should be good medicines in the school, so if girls are menstruating and feel a lot of pain they can be given these medications to help encourage girls to come to school” *

*(Post-menarche girls-FGD4)*



*“The toilets should have functional taps, also soap and water should be in the toilets instead of having to come to the classroom to wash hands” *

*(Post-menarche girls-FGD9)*



*“There should be water in the toilet, most schools at the moment have taps outside, not so close to the toilet, so you need to fetch water and take it inside before you go.” *

*(Post-menarche girls-FGD5)*


#### 3.4.2. Quantitative Findings

Half of the participants (51%) reported missing at least one school day in the last month. The most frequent reasons reported for missing school in the last month were illness (31%), menstruation (27%), and domestic duties (22%) (Table 5). The main reasons reported for missing school while menstruating were menstrual pain (33%) and being afraid of staining clothes with menstrual blood (15%) (Table 5). Fear of body odour was mentioned frequently in interviews with girls, however, only 2% of girls surveyed reported that as a reason for missing school (Table 5). The survey showed 82% of the participants experienced some level of pain; girls in the Arabic school were slightly more likely to report experiencing pain than girls in English-based schools (89% vs. 80%; *p* = 0.047) (Table 5). Univariable analysis showed that the odds of missing school increased with higher levels of pain experienced, and this association was maintained even after adjusting for potential confounders (mild pain: adjusted odds ratio (AOR) 2.36 (95% CI: 1.02–5.48) and extreme pain AOR 16.8 (95% CI: 7.28–38.74)) (Table 6).

Increase in age seemed to increase the odds of missing school while menstruating (15–17 years odds ratio (OR) = 4.72 (95% CI: 1.42–15.67) and (18–25 years OR = 5.42 (95% CI: 1.60–18.40)). The type of school the girls went to did not influence the odds of them missing school. The girls whose mother had Arabic education as opposed to no education reduced the odds of missing school OR = 0.43 (95% CI: 0.28–0.66), however, fathers’ education level had no consistent influence (Table 6).

Using a combination of reusable cloth and disposable pads almost doubled the odds of missing school compared with using just reusable absorbents, this association was maintained when adjusting for confounders (AOR = 1.68 (95% CI: 1.05–2.69)). There was no strong difference in school attendance if the absorbent material used was reusable cloth vs. disposable pads. Not being able to buy pads increased the odds of missing school (Unadjusted OR 1.60 (95% CI: 1.06–2.40)), the effect was attenuated somewhat after adjusting for potential confounders (AOR 1.41 (95% CI: 0.92–2.16)) (Table 6). The school giving pads did not affect the odds of missing school.

Being happy to use the school latrines while menstruating strongly reduced the odds of missing school, this association was maintained even after adjusting for confounders (AOR 0.59 (95% CI: 0.37–0.93)). Not having gender specific toilets strongly increased the odds of missing school (unadjusted OR = 2.71 (95% CI: 1.90–3.88)), however, a school not having gender specific toilets was highly associated with school type, where only one Arabic school was found to not have gender specific toilets. Having soap available for handwashing almost halved the odds of missing school while menstruating (AOR = 0.46 (95% CI: 0.3–0.73)). Having clean toilets strongly reduced the odds of missing school even after adjusting for confounders (AOR = 0.44 (95% CI: 0.26–0.75)). However, having privacy in the school toilets, water in at least one toilet cubicle, or the frequency of changing menstrual absorbents did not strongly affect the odds of missing school among the girls surveyed (Table 6).

Learning about menstruation before menarche did not affect the odds of missing school while menstruating (AOR 0.92 (95% CI: 0.62–1.34)), similarly, having good knowledge about menstruation was not seen to protect against missing school (AOR 0.98 (95% CI: 0.65–1.48)) (Table 6).

Having at least one RTI symptom almost doubled the odds of missing school while menstruating (AOR = 1.99 (95% CI: 1.34–2.94)). Likewise, at least one UTI symptom also increased the odds of missing school (AOR = 1.71 (95% CI: 1.16–2.52)) (Table 6).

The self-completed diaries did not work in this context, there were many errors in completion (in some cases all boxes on the page were checked, some indicated they attended school seven-days a week even though this was not the case, some indicated they were menstruating the whole month, or selected not menstruating and heavy menstruation for the same day, in some cases all girls in the class had completed the diary in the exact same way) or diaries were incomplete. Furthermore, some teachers reported helping the students fill in the diaries, which can reduce effectiveness of the diaries as this makes them less private, and girls may hide the true information from the teachers. Therefore, this data had to be discarded and could not be compared with the absenteeism data from the survey.

### 3.5. Exploration of Reasons Related to Girls Drop Out

#### Qualitative Findings

The most common factors for leaving school were lack of money, challenging teacher–student relationships, burden of domestic duties, marriage, and pregnancy. Reports suggested that if marriage was not the reason for leaving school, girls were married soon after being taken out of school, regardless of the age of leaving school.

One of the reasons reported for leaving school was lack of money and prioritising the male child’s education over that of the female child.


*“(I left school) Due to lack of money, we couldn’t afford to pay the school fees. My parents work on the farm, when they sell the produce, they use that to pay the fees, but we were many children going to school and it was difficult for them to pay for all of us, so my father decided to take me out of school and let my elder brothers continue…when I left school (at 17 years) I got married… even now if I had the chance, I would be willing to learn” *

*(IDI-dropout1)*


Another factor that leads to dropping out was the teachers’ approach to the students.


*“When I was in school the teachers used to beat us very seriously if we did something wrong. The will just straighten us on the table and beat us on our buttocks…that could lead that girl to go out of school completely, that is not good and should be stopped.” *

*(IDI-dropout1)*


The burden of domestic duties was another factor for girls dropping out of school.


*“From school I should be studying, but instead I do household duties… sometimes if I come home for break from school, I don’t go back to school that day because of the housework” *

*(IDI-dropout5)*


None of the interviews with girls who had dropped out of school listed menstruation as a factor leading to leaving school. Many girls interviewed felt that menstruation can make you miss school, but not drop out, “*Pain during menstruation can make girls miss a few days of school, but it cannot be a reason for them to leave school completely”* (post-menarche girls-FGD8). However, others mentioned cultural beliefs or practices linked to menarche that could be a cause of dropping out “*For some, once they see their menses… they will be taken out of school to them married”* (post-menarche girls-FGD6).

## 4. Discussion

This study shows that the management of menstruation in rural schools of The Gambia is a real challenge for schoolgirls, and it is clearly associated with school attendance. Both qualitative and quantitative findings show that menstrual pain, inadequate school WASH facilities, cultural beliefs of restricted movement or fear of people ascertaining their menstrual status, or symptoms of urogenital infection were the main drivers for school absenteeism while menstruating.

These findings support previous studies in LMICs that show menstruation is a barrier to school attendance [25,27,46,47]. The results suggest 27% of girls missed at least one day of school per month during their period, which is substantial and in line with proportions of absenteeism that have been reported by other studies in West African contexts [10]. It is important to note there are challenges in accurately measuring school attendance [5,12,25,48,49], with retrospective surveys being more likely to underreport due to social desirability bias or recall bias. In The Gambia, menstruation is often considered an illness, therefore, in this study, when capturing information on reasons for missing school, the questions were broken down to separate illness and menstruation, however, it is possible that menstrual-related absenteeism could have been underreported and reported as illness instead. Additionally, school attendance is difficult to measure accurately using school registers as they are frequently incomplete, inaccurate, or girls use different names for school enrolment and study enrolment [50,51,52]. A study in Uganda found the use of daily self-completed diaries to be useful and a way to validate attendance data [25], however, in this context this method was ineffective. The inaccuracies and incomplete diaries could have been due to poor literacy levels, not understanding how to complete or forgetting to complete the diaries. Indeed, further work is needed to validate the accuracy of tools measuring school absenteeism, or create new methods to measure this data accurately.

In this study, most girls did not go to school because of menstrual pain, and similar findings have been seen in other contexts as well [12,22,23]. Evidence of the impact of pain medication on absenteeism is limited [5], taboos and myths of using pain medication during menstruation exist in some contexts [25,53] and should be explored in the Gambian context when designing interventions to address pain management. Informal discussions with women in the communities have suggested there may be thoughts that pain medication can lead to infertility.

Several studies, including this one, have also found that missing school while menstruating is associated with older age [9,25]. The reasons why this is the case are not well understood but could be that older girls may experience more teasing/shame or be more affected by it, or they could also be at a higher risk of urogenital infections. Specific causes of menstruation-related absenteeism among older girls vs. younger girls should be investigated further.

Interviews with the students highlighted that girls were missing school due to fear of being teased (by male and female students or male staff members), or fear of others finding out they were menstruating, this is seen in many contexts [25,54,55,56], and the latter is often linked to cultural beliefs that menstruation should be hidden. This highlights the importance of involving multiple different groups in the community, especially the elders and male members in discussions about puberty and menstruation to help improve knowledge and break these beliefs and taboos. It is also important to note that these factors can also affect participation, engagement, and performance in school [24,57].

School WASH facilities were seen to be important to adequate menstrual hygiene management practices and school attendance. Access to soap and water in toilet areas and clean toilets were the main factors associated with reduced school attendance in our study. Not having gender specific toilets was strongly associated with school attendance in crude analysis. Because of the strong co-linearity between gender specific toilets and school type, the scope of the multivariable analysis was limited, making it difficult to establish a causal link, therefore, this aspect requires further study. Other studies also found having privacy at school toilets (doors and locks, away from prying eyes) were important [55,56,58]. The prying eyes concept could be linked to gender specific toilets, although it was not specifically stated as such, however, this study did not find having doors and locks as being associated with school absenteeism. This difference could be because a higher percentage of girls had access to at least one toilet cubicle with a door, or in the local Gambian context, it was noted that girls usually go to the toilet in pairs, so they may feel having a friend nearby can deter others from coming and hence act as a mode of increasing privacy.

Our study showed that there was no major difference in prevalence of absenteeism if the girls used reusable cloths or disposable pads. This is contrary to many other studies, which have shown that using disposable sanitary pads results in lower prevalence of absenteeism [12,56]. The data does suggest that using both types of absorbent material did increase prevalence of absenteeism. The reasons for this are unclear and need to be explored, but current speculations could be that when the preferred material is finished the girls are less likely to want to go out in the less-preferred material, or using both types of material could result in an inappropriate use of one or the other material, as both material types have different usage strategies and therefore inappropriate use and maintenance can increase leaking or adverse health outcomes such as urogenital infections [59], thus leading to school absenteeism. The effects of urogenital infections on school absenteeism have not been well explored; this study found that girls that reported symptoms of urinary and reproductive tract infections were more likely to miss school, which could be due to a higher level of discomfort and pain. It is suggested that a study with a clinical aspect that can test accurately for urogenital infections could be useful in establishing linkages.

The MoBSE in The Gambia reported [33] preliminary findings from a cross-sectional survey that showed girls who attended a school where they piloted distribution of disposable sanitary pads had improved school attendance from 68% to 90%. However, in our study, we found attending a school that supplied disposable sanitary pads was not associated with reducing school absenteeism. This could be partly explained by the mismatch between the number of disposable pads supplied by the schools and the actual numbers needed by the girls during their menstrual period, or that girls were not taught how to use the pads properly. A more detailed evaluation of the disposable sanitary pads’ distribution should be carried out to assess its impact on attendance while menstruating. Inability to buy disposable sanitary pads was associated with missing school, the effect was diluted after adjusting for potential confounders but remains strong enough to merit further study.

Mothers’ education was associated with school absenteeism, which is in line with other studies [12,60]. This is likely to be because mothers are among the main point of contact for girls about menstruation and how to handle themselves while menstruating, therefore, it is important to ensure mothers have adequate knowledge about menstruation and are able and comfortable to share it with their children.

This study has some limitations. The survey had to be administered rather than self-completed which could add in some biases, however, this approach was the only feasible option due to points mentioned earlier, and it may also have had the benefit that enumerators could explain any question that the respondents did not understand. Another limitation is the results of school attendance only rely on self-reported school attendance data which is subject to recall and reporting bias. It was not possible to compare attendance data to the diary data due to errors in completion, therefore, other ways to verify attendance data should be explored. Additionally, this study did not capture data on performance and engagement in school while menstruating, and it will be important to include in further studies. This can be performed through self-reported perceptions on participation, engagement, and using school examination data. The study does use qualitative data collection to explore many aspects of menstrual experience and its effect on school attendance, the results from this are less replicable and verifiable, can be open to bias in how the moderator asks question, or can be subjective during analysis; however, we did try to mitigate this by ensuring all moderators had sufficient training to ensure leading questions were not used and non-judgmental responses were given. A subset of the analysis was carried out by two of the authors to reduce chances of subjective analysis.

## 5. Conclusions

Menstruation was associated with reduced school attendance among adolescent girls in certain rural Gambian communities. Menstrual pain, school WASH facilities, menstrual absorbent access, and urogenital infections were the key factors that influenced school attendance during menstruation. The results also show that cultural norms limited access to knowledge and information on optimal MHM practices among the adolescent girls interviewed.

The high rate of school absenteeism due to menstruation is cause for alarm and requires attention. The findings from this study suggest that the main target areas include improving communications about menstruation among community members and including boys and men in these conversations to challenge cultural misconceptions and taboos surrounding menstruation. The study findings also highlight that mothers and teachers could benefit from trainings to improve their understanding of menstruation in order to teach the younger generations. More work is needed to create content that can be added into the school curriculum to improve knowledge of menstruation among adolescents.

Menstrual pain reduction or management is another key area to consider for intervention development. These strategies can be included in school adolescent programmes or added to the school policies for first aid measures.

Acceptable and sustainable school WASH facility improvements (including better access to water, water closer to the toilets/inside the toilets, availability of soap, and clean toilet facilities) should be discussed with the community and ministries of health, education, and water and sanitation to create new school WASH facility guidelines.

Menstrual absorbents are also another area that needs focus; further information about preference of absorbent material, how to improve access to absorbent materials, and if adolescents know how to use the material adequately are essential to help reduce urogenital infections and prevent girls from restricting movement while menstruating. Discussing different and/or more sustainable absorbents could also be a priority.

The Gambia has made great strides in increasing female students’ enrolment in school, however, attendance, attention/participation in class, and progressing to higher levels of education has been a challenge; addressing these concerns linked to menstrual health can help reduce these challenges and help continue the strides the country is making in gender equity and empowerment.

There is a lack of robust evidence of the effectiveness of MHM interventions to improve school attendance; therefore, we propose all these aspects to be combined into a multicomponent intervention package to generate evidence to inform policies.

## Figures and Tables

**Figure 1 ijerph-19-03337-f001:**
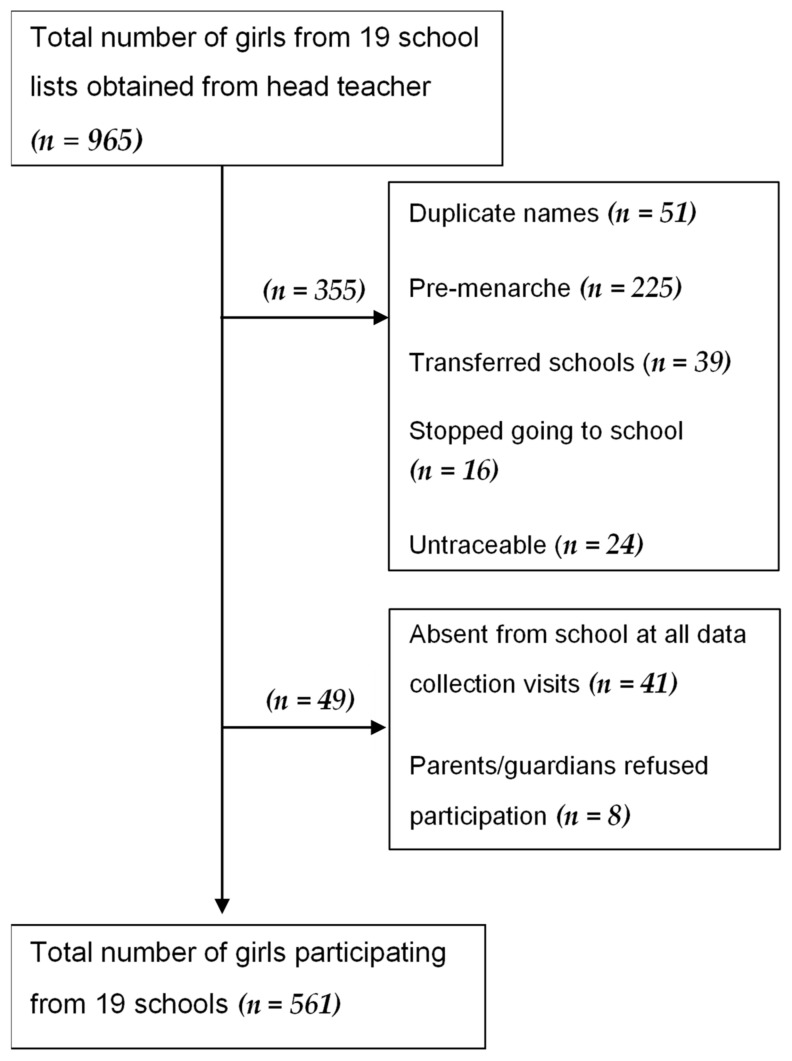
Flow diagram of sampling process.

**Table 1 ijerph-19-03337-t001:** Number of qualitative discussions and participants.

Tools	Participants	Number of Groups	Total Number of Participants	Mean Age of Participants (Years)
FGDs	Post-menarche girls	9	64	15.7
	Pre-menarche girls	5	32	13.2
	Boys	6	38	16.7
IDIs	Drop-out girls	6	6	17.5
	Teachers	3	3	
	Mothers	7	7	
	Boys	5	5	
	Total	41	155	

**Table 2 ijerph-19-03337-t002:** Sociodemographic characteristics of study participants.

Socioeconomic Characteristics	Overall *n* = 561 (%)	English *n* = 402 (%) (12 Schools)	Arabic *n* = 159 (%) (7 Schools)	Difference between School Type
Age				*p* = 0.713
11–14	40 (7.14)	33 (8.21)	7 (4.40)	
15–17	354 (63.04)	251 (62.44)	103 (64.78)	
18–25	167 (29.82)	118 (29.35)	49 (30.82)	
Grade				*p* = 0.13
5–6	54 (9.63)	22 (5.47)	32 (20.13)	
7–9	336 (59.89)	233 (57.96)	103 (64.78)	
10–12	171 (30.48)	147 (36.57)	24 (15.09)	
Muslim	561 (100)	402 (100)	159 (100)	–
Transportation to school				*p* = 0.002
Walk to school	472 (84.14)	317 (78.86)	155 (97.48)	
Cycle to School	89 (15.86)	85 (21.14)	4 (2.52)	
Mothers’ education				*p* = 0.003
No Formal	175 (31.19)	142 (35.32)	33 (20.75)	
Arabic	257 (45.81)	177 (44.03)	80 (50.31)	
Primary	41 (7.31)	32 (7.96)	9 (5.66)	
Secondary	27 (4.81)	23 (5.72)	4 (2.52)	
Do not know	61 (10.87)	28 (6.97)	33 (20.75)	
Fathers’ education				*p* = 0.172
No Formal	88 (15.69)	74 (18.41)	14 (8.81)	
Arabic	310 (55.26)	212 (52.74)	98 (61.64)	
Primary	18 (3.21)	15 (3.73)	3 (1.89)	
Secondary	63 (11.23)	49 (12.19)	14 (8.81)	
Do not know	82 (14.62)	52 (12.94)	30 (18.87)	
Source of income				*p* = 0.122
Farming	378 (67.38)	291 (72.39)	87 (54.72)	
Business	50 (8.91)	39 (9.70)	11 (6.92)	
Salary	90 (16.04)	55 (13.68)	35 (22.01)	
Trading	27 (4.81)	8 (1.99)	19 (11.95)	
Other	16 (2.85)	9 (2.24)	7 (4.40)	
Water Source at home				*p* = 0.021
Unprotected well	29 (5.17)	29 (7.21)	0 (0)	
Protected Well	8 (1.43)	8 (1.99)	0 (0)	
Community hand pump	115 (20.50)	109 (27.11)	6 (3.77)	
Community standpipe	385 (68.96)	232 (57.71)	153 (96.23)	
Household water	24 (4.28)	24 (5.97)	0 (0)	
Time to collect water at home				*p* = 0.16
Less than 30 min to get water	467 (83.24)	321 (79.85)	146 (91.82)	
More than 30 min to get water	94 (16.76)	81 (20.15)	13 (8.18)	
Type of toilet facility at home				*p* = 0.21
Pit latrine without a slab	71 (12.66)	57 (14.18)	14 (8.81)	
Pit latrine with a slab	470 (83.78)	333 (82.84)	137 (86.16)	
Pour flush or flush toilet	20 (3.57)	12 (2.99)	8 (5.03)	
Wealth Index				*p* = 0.236
Low	188 (33.51)	152 (37.81)	36 (22.64)	
Middle	193 (34.40)	131 (32.59)	62 (38.99)	
High	180 (32.09)	119 (29.60)	61 (38.36)	

**Table 3 ijerph-19-03337-t003:** Menstrual knowledge, practices, attitudes, and urogenital symptoms of study participants.

**Knowledge and Attitudes about Menstruation**	**Overall *n* = 561 (%)**	**English *n* = 402 (%) (12 Schools)**	**Arabic *n* = 159 (%) (7 Schools)**	**Difference between School Types**
Average age of menarche	14.22 (1.29)	14.31 (1.21)	14.01 (1.45)	*p* = 0.354
Learn about menstruation before menarche	340 (60.61)	254 (63.18)	148 (36.82)	*p* = 0.07
Average age of learning about menstruation ^+^	13.57 (1.70)	13.53 (1.69)	13.68 (1.72)	*p* = 0.708
Knowledge questions				
Knew old women do not menstruate *	403 (71.84)	305 (75.87)	98 (61.64)	*p* = 0.092
Knew menstruation was not a disease *	307 (54.72)	223 (55.47)	84 (52.83)	*p* = 0.821
Knew pregnant women do not menstruate *	466 (83.07)	344 (85.57)	122 (76.73)	*p* = 0.031
Knew menstrual blood does not come from the stomach *	278 (49.55)	212 (52.74)	66 (41.51)	*p* = 0.145
Knew menstrual blood comes from the uterus *	273 (48.66)	217 (53.98)	56 (35.22)	*p* = 0.009
Good Knowledge score (uses *)	366 (65.24)	280 (69.65)	86 (54.09)	*p* = 0.009
Menstrual hygiene management practices				
Absorbent Material used				*p* = 0.661
Disposable pads	117 (20.86)	112 (27.86)	5 (3.14)	
Reusable cloth	196 (34.94)	101 (25.12)	95 (59.75)	
Combination of both disposable and reusable materials	248 (44.21)	189 (47.01)	59 (37.11)	
Unable to buy disposable pads	146 (26.02)	89 (22.14)	57 (35.85)	*p* = 0.016
Pads given by school				
Yes	337 (60.07)	334 (83.08)	3 (1.89)	*p* ≤ 0.001
Number of pads given (10 units per pack)				*p* ≤ 0.001
0/did not go to collect	39 (11.61)	38 (11.41)	1 (33.33)	
1 pack	165 (47.11)	163 (48.95)	2 (66.67)	
2 packs	127 (37.80)	127 (38.14)	0 (0)	
3 packs or more	5 (1.49)	5 (1.50)	0 (0)	
Frequency of change on heavy bleeding day				*p* = 0.029
Once a Day	20 (3.57)	10 (2.49)	10 (6.29)	
Twice a day	221 (39.39)	149 (37.06)	72 (45.28)	
Three or more times a day	320 (57.04)	243 (60.45)	77 (48.43)	
How the reusable material is washed at home	*n = 446*	*n = 291*	*n = 155*	*p* = 0.047
With Water	15 (3.36)	7 (2.41)	8 (5.16)	
With water and soap/detergent	430 (96.41)	283 (97.25)	147 (94.84)	
How the reusable material is dried				*p* = 0.536
Outside (open bathroom or sun)	362 (81.72)	238 (82.64)	124 (80.00)	
Inside house or under mattress	81 (18.28)	50 (17.36)	31 (20.00)	
**Urinary Tract Infections and Reproductive Tract Infections**	**Overall *n* = 561 (%)**	**English *n* = 402 (%) (12 Schools)**	**Arabic *n* = 159 (%) (7 Schools)**	**Difference between School Types**
Urinary Tract Infection (UTI) Symptoms				
Feeling of burning and discomfort while urinating	124 (22.10)	90 (22.39)	34 (21.38)	*p* = 0.894
Had to wake up and pass urine more than usual	56 (9.98)	36 (8.96)	20 (12.58)	*p* = 0.433
Cloudy urine or blood in your urine	131 (23.35)	96 (23.88)	35 (22.01)	*p* = 0.751
Symptoms indicative of UTI (at least 1 symptom)	221 (39.39)	161 (40.05)	60 (37.74)	
Reproductive Tract Infections (RTI) Symptoms				
Abnormal vaginal discharge	119 (21.21)	84 (20.90)	35 (22.01)	*p* = 0.790
Feeling of burning or itching in the genitals	206 (36.72)	141 (35.07)	65 (40.88)	*p* = 0.412
Symptoms indicative of RTI (at least 1 symptom)	263 (46.88)	180 (44.78)	83 (52.20)	

^+^ 11 Girls reported not having learnt about menstruation at the time of the survey. * Questions used to generate knowledge score.

**Table 4 ijerph-19-03337-t004:** Access to water and sanitation at school.

**Access to Water and Sanitation at School (Spot Check Data)**	**Overall *n* = 19 (%)**	**English *n* = 12 (%)**	**Arabic *n* = 7 (%)**	**Difference between School Types**
Toilet facilities				
Gender Specific Toilets	18 (94.74)	12 (100)	6 (85.71)	*p* = 0.179
Clean (at least 50% of the cubicles were clean)	17 (89.47)	11 (91.67)	6 (85.71)	*p* = 0.683
Privacy				
Door	15 (78.95)	12 (100)	3 (42.86)	*p* = 0.003
Lockable Door	13 (68.42)	12 (100)	1 (14.29)	*p* ≤ 0.001
Water in at least one toilet cubicle	4 (21.05)	2 (16.67)	2 (28.57)	*p* = 0.539
Type of handwashing facilities in schools				
Running water from a piped system or tank	10 (53.63)	7 (58.33)	3 (42.86)	*p* = 0.515
Hand-poured water system (bucket or ladle)	16 (84.21)	9 (75)	1 (100)	*p* = 0.149
Availability of soap for handwashing	7 (36.84)	6 (50)	1 (14.29)	*p* = 0.12
	**Overall *n* = 561 (%)**	**English *n* = 402 (%) (12 schools)**	**Arabic *n* = 159 (%) (7 Schools)**	**Difference between School Types**
How girls feel using the school latrines while menstruating				*p* = 0.003
Unhappy	296 (62.05)	207 (57.50)	89 (76.07)	
Happy	181 (37.95)	153 (42.50)	28 (23.93)	

**Table 5 ijerph-19-03337-t005:** Missing School data.

Missing School	Overall *n* = 561 (%)	English *n* = 402 (%) (12 Schools)	Arabic *n* = 159 (%) (7 Schools)	Difference between School Types
Absenteeism				
Missed 1 day or more of school last month	286 (50.98)	200 (49.75)	86 (54.09)	*p* = 0.731
Missed 1 day or more because of illness	175 (31.19)	113 (28.11)	62 (38.99)	*p* = 0.072
Missed 1 day or more because of lack of money	85 (15.15)	37 (9.20)	48 (30.19)	*p* = 0.005
Missed 1 day or more because of domestic duties	123 (21.93)	92 (22.89)	31 (19.50)	*p* = 0.586
Missed 1 day or more because of menstruation	152 (27.09)	107 (26.62)	45 (28.30)	*p* = 0.764
Reasons for missing school linked to menstruation (mark all that applies)				
Afraid of staining clothes	82 (14.62)	63 (15.67)	19 (11.95)	*p* = 0.534
Afraid of bad smell	13 (2.32)	10 (2.49)	3 (1.89)	*p* = 0.79
Pain	184 (32.80)	129 (32.09)	55 (34.59)	*p* = 0.546
Feeling uncomfortable and tired	38 (6.77)	29 (7.21)	9 (5.66)	*p* = 0.510
Having no place to wash/change in school	10 (1.78)	6 (1.49)	4 (2.52)	*p* = 0.527
Teasing	11 (1.96)	6 (1.49)	5 (3.14)	*p* = 0.40
Average pain				*p* = 0.047
No Pain	99 (17.65)	82 (20.40)	17 (10.69)	
Mild pain	283 (50.45)	198 (49.25)	85 (53.46)	
Extreme pain	179 (31.91)	122 (30.35)	57 (35.85)	

**Table 6 ijerph-19-03337-t006:** Analysis of factors associated with missing at least one day of school due to menstruation.

**Socioeconomic Characteristics**	** *n* **	**Number Missing at Least One Day of School Due to Menstruation (%)**	**Odds Ratio (OR)**	***p*-Value**	**Adjusted OR (AOR)**
Total	561	152 (27.09)			
Age					
10–14	40	3 (7.50)	1		
15–17	354	98 (27.68)	4.72 (1.42–15.67)	*p* = 0.011	
18–25	167	51 (30.54)	5.42 (1.60–18.40)	*p* = 0.007	
Grade					
4–8	54	12 (22.22)	1		
9–11	336	86 (25.60)	1.20 (0.61–2.40)	*p* = 0.596	
12	171	54 (31.58)	1.62 (0.79–3.31)	*p* = 0.191	
Type of school					
English	402	107 (26.62)	1		
Arabic	159	45 (28.30)	1.09 (0.72–1.64)	*p* = 0.686	
Way to reach school					
Walking	472	125 (26.48)	1		
Cycling	89	27 (30.34)	1.21 (0.74–1.99)	*p* =0.45	
Mother education					
No Formal	175	63 (36.00)	1		
Arabic	257	50 (19.46)	0.43 (0.28–0.66)	*p* ≤ 0.001	
Primary	41	16 (39.02)	1.14 (0.57–2.29)	*p* = 0.718	
Secondary	27	8 (29.63)	0.75 (0.31–1.81)	*p* = 0.52	
Do not know	61	15 (24.59)	0.58 (0.30–1.12)	*p* = 0.105	
Father Education					
No Formal	88	29 (32.95)	1		
Arabic	310	76 (24.52)	0.66 (0.40–1.11)	*p* = 0.114	
Primary	18	7 (38.89)	1.29 (0.45–3.69)	*p* = 0.629	
Secondary	63	13 (20.63)	0.53 (0.25–1.13)	*p* = 0.098	
Do not know	82	27 (32.93)	1.00 (0.53–1.89)	*p* = 0.997	
Wealth Index					
Low	188	44 (23.40)	1		
Middle	193	53 (27.46)	1.24 (0.78–1.97)	*p* = 0.364	
High	180	55 (30.56)	1.44 (0.91–2.29)	*p* = 0.123	
**Menstrual Hygiene Management**	** *n* **	**Number Missing at Least One Day of School Due to Menstruation (%)**	**Odds Ratio (OR)**	***p*-Value**	**Adjusted OR (AOR) ****
Absorbent Material normally used					
Reusable cloth	196	41 (20.92)	1		
Disposable pads	117	28 (23.93)	1.19 (0.69–2.05)	*p* = 0.53	1.31 (0.71–2.39)
Combination of both reusable and disposable pads	248	83 (33.47)	1.90 (1.23–2.93)	*p* = 0.004	1.68 (1.05–2.69)
Being unable to buy disposable pads despite wanting to					
No	415	102 (24.58)	1		
Yes	146	50 (34.25)	1.60 (1.06–2.40)	*p* = 0.024	1.41 (0.92–2.16)
School giving pads for free					
No	224	55 (24.55)	1		
Yes	337	97 (28.78)	1.24 (0.84–1.83)	*p* = 0.27	1.73 (0.87–3.44)
How girls feel using the latrine at school whilst menstruating					
Unhappy	296	91 (30.74)	1		
Happy	181	39 (21.55)	0.62 (0.40–0.95)	*p* = 0.029	0.59 (0.37–0.93)
Frequency of change menstrual absorbent on heavy bleeding days					
Less than three times a day	241	63 (26.14)	1		
Three times a day or more	320	89 (27.81)	1.09 (0.75–1.59)	*p* = 0.659	1.13 (0.80–1.61)
How the reusable material is washed at home					
With Water	15	2 (13.33)	1		
With water and soap/detergent	430	122 (28.37)	2.57 (0.57–11.58)	*p* = 0.218	1.58 (0.42–5.95)
How the reusable material is dried					
Outside (open bathroom or sun)	362	107 (29.56)	1		
Inside house or under mattress	81	17 (20.99)	0.63 (0.35–1.13)	*p* = 0.12	0.66 (0.36–1.21)
When they learnt about menstruation					
After menarche	221	61 (27.60)	1		
Before menarche	340	91 (26.76)	0.96 (0.66–1.40)	*p* = 0.827	0.92 (0.62–1.36)
Knowledge score					
Poor	195	51 (26.15)	1		
Good	366	101 (27.60)	1.08 (0.73–1.60)	*p* = 0.714	0.98 (0.65–1.48)
**Health Symptoms**	** *n* **	**Number Missing at Least One Day of School Due to Menstruation (%)**	**Odds Ratio (OR)**	***p*-Value**	**Adjusted OR (AOR) ****
Having at least one UTI symptom					
No	340	77 (22.65)	1		
Yes	221	75 (33.94)	1.75 (1.20–2.56)	*p* = 0.003	1.71 (1.16–2.52)
Having at least one RTI symptom					
No	298	60 (20.13)	1		
Yes	263	92 (34.98)	2.13 (1.46–3.12)	*p* ≤ 0.001	1.99 (1.34–2.94)
Average pain					
No Pain	99	7 (7.07)	1		
Mild pain	283	43 (15.19)	2.35 (1.02–5.42)	*p* = 0.044	2.36 (1.02–5.48)
Extreme pain	179	102 (56.98)	17.41 (7.64–39.67)	*p* ≤ 0.001	16.8 (7.29–38.74)
**WASH Questions from School**	** *n* **	**Number Missing at Least One Day of School Due to Menstruation (%)**	**Odds Ratio (OR)**	***p*-Value**	**Adjusted OR (AOR) ****
Gender Specific toilets					
Yes	557	150 (26.93)	1		
No	4	2 (50)	2.71 (1.90–3.88)	*p* ≤ 0.001	2.25 (0.29–17.60)
At least half the cubicles are clean					
No	75	31 (41.33)	1		
Yes	486	121 (24.90)	0.47 (0.26–0.86)	*p* = 0.014	0.44 (0.26–0.75)
Door on the toilets					
No	72	20 (27.78)	1		
Yes	489	132 (26.99)	0.96 (0.56–1.64)	*p* = 0.885	0.77 (0.37–1.59)
Door that is lockable					
No	105	32 (30.48)	1		
Yes	456	120 (26.32)	0.81 (0.48–1.38)	*p* = 0.447	0.53 (0.24–1.17)
Water in at least one of the toilet cubicles					
No	475	122 (25.68)	1		
Yes	86	30 (34.88)	1.55 (0.96–2.50)	*p* = 0.072	1.20 (0.71–2.00)
Soap available for handwashing					
No	319	110 (34.48)	1		
Yes	242	42 (17.36)	0.40 (0.23–0.68)	*p* = 0.001	0.46 (0.3–0.73)

** age, wealth index, school type, and maternal education were used to adjust for confounding.

## Data Availability

The datasets used and/or analysed during the current study are available from the corresponding author on reasonable request.

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
