# Peer review of "Effects of Menstrual Health and Hygiene on School Absenteeism and Drop-Out among Adolescent Girls in Rural Gambia"

_ijerph, 2022, doi:10.3390/ijerph19063337_

Round 1

Reviewer 1 Report

Manuscript ID: ijerph-1601249

Title: "Effects of menstrual health and hygiene on school absenteeism and drop-out among adolescent girls in rural Gambia"

FIRST REVIEW REPORT

In the paper, the Authors explored how menstrual hygiene management practices and related factors influence school absenteeism and drop-out among primary and secondary school girls in rural Gambia.

The topic of the manuscript is interesting as well as the potential academic contribution of the work, but the Authors should improve their research according to the following indications.

1. In the introduction section:

1.1. It should be discussed international situation, regulations, and approaches, and should be motivated the research to be of high interest for a even broader group of addressees.

1.2. It should be explained how the article has been structured by presenting the different sections.

2. Literature review is poor and should be extended.

3. Implications, limitations of the study and future research directions should be addressed.

4. Conclusions section is too brief and should be expanded.

5. Tables and figure should report the sources.

6. An extensive editing of English language and style is required.

Author Response

Dear Reviewer,

Thank you for your time reviewing the manuscript and providing valuable feedback. 

  1. In the introduction section:

1.1. It should be discussed international situation, regulations, and approaches, and should be motivated the research to be of high interest for a even broader group of addressees.

Answer: We have added more information about the international situation, factors that can be linked to absenteeism and the implications of this, showing why addressing this important topic is relevant for researchers and practitioners. The whole introduction section has been redone.

1.2. It should be explained how the article has been structured by presenting the different sections.

Answer: At the end of the introduction we have presented the objectives of the study, which will structure the way the result section is presented.

  1. Literature review is poor and should be extended.

Answer: We had originally kept the introduction section quite short as we felt the paper was getting too long and may have been too much information for a reader, but we have now added some more information for the literature review in the introduction section.

  1. Implications, limitations of the study and future research directions should be addressed.

Answer: Limitations were added throughout the discussion for each section that was appropriate, however we have now added a separate paragraph for limitations at the end of the discussion. Implications and further research have been added to the conclusion.

  1. Conclusions section is too brief and should be expanded.

Answer: We have added more information to the conclusion section

  1. Tables and figure should report the sources.

Answer: The figures and tables were created by the authors of this paper, so there are no sources.

  1. An extensive editing of English language and style is required.

Answer: We have edited the document and shared with a native English speaker to help improve language and style.

Reviewer 2 Report

Good work. It was a pleasure to read such a work

Author Response

Thank you for reviewing our manuscript.

Reviewer 3 Report

This review is interesting and important for understanding effects of menstrual health and hygiene on school absenteeism and drop-out among adolescent girls in rural Gambia. But several points must be improved.

(Comment 1) Authors commented that "Menstruation was associated with reduced school attendance among adolescent girls in these rural Gambian communities". (line 516-517) Regarding this, I recommend authors to supplement subtitle "1.1. Literature review" under the "1. Introduction".

(Comment 2) The authors used surveys, qualitative interviews, daily diaries, and unannounced water sanitation and hygiene (WASH) spot checks. Thus, detail method for each item must be presented. I recommend authors to re-write "Materials and Methods Section". Regarding unannounced water sanitation and hygiene (WASH) spot checks, what is criteria for this? is it international standard? or derived from previous research? This part should be written in detail at the same level as surveys and qualitative interviews.

e.g.

Materials and Methods

(https://doi.org/10.3390/healthcare9050493)

2.1. Method for survey

2.1.1. Survey sampling

2.1.2. Development of the survey form

2.1.3. Survey distribution

2.1.4. Data collection and analysis

(https://doi.org/10.3390/ijerph19031739)

2.2. Qualitative interviews

2.1.1. Study Participants

2.1.2. Development of the questionnaire

2.1.3. Interview process

2.1.4. Data collection and analysis

2.3. water sanitation

2.4. hygiene (WASH) spot checks

(Comment 3) The mixed method has considerable difficulty in understanding the paper. I consider this to be a limitation of this study. Thus, I recommend authors to supplement study limitation(e.g. mixed method) in "Discussion section".

Author Response

Dear Reviewer,

Thank you for your time reviewing the manuscript and providing valuable feedback.

Comment 1: Authors commented that "Menstruation was associated with reduced school attendance among adolescent girls in these rural Gambian communities". (line 516-517) Regarding this, I recommend authors to supplement subtitle "1.1. Literature review" under the "1. Introduction".

Answer: We have added more information from the literature review conducted to give the reader more context, but have opted to make it flow throughout the introduction section rather than create a new section for it.

 Comment 2: The authors used surveys, qualitative interviews, daily diaries, and unannounced water sanitation and hygiene (WASH) spot checks. Thus, detail method for each item must be presented. I recommend authors to re-write "Materials and Methods Section".

Answer: We have added more details to the methods section and have rewritten some of the information presented, however we did not completely separate the sections as you suggested, as we felt that lead to a lot of repetition, therefore we opted to separate the sections using some of your suggestion just for the study design, data collection and data analysis sections. We hope now it is more clear for readers.

Comment 2.1: Regarding unannounced water sanitation and hygiene (WASH) spot checks, what is criteria for this? is it international standard? or derived from previous research? This part should be written in detail at the same level as surveys and qualitative interviews.

Answer: We have added more information in the paper (there is a WASH spot check subheading under each of the main methods section; study design and sampling, data collection and data management and analysis,). We have also described that the WASH tools used has been adapted from the UNICEF tools used to assess WASH facilities in schools. We also included some information about how we piloted the WASH tool to adapt it into our context (section 2.4.5).

 Comment 3: The mixed method has considerable difficulty in understanding the paper. I consider this to be a limitation of this study. Thus, I recommend authors to supplement study limitation (e.g. mixed method) in "Discussion section".

Answer: When designing the study, we opted for a mixed methods approach because combining the two type of data allow us to benefit from both the detailed contextualized insight of qualitative data and the generalizable, externally valid insights of quantitative data. Triangulation of methods can provide opportunities for testing alternative interpretations of data and for examining the extent to which the context helped to shape the quantitative results.
We also recognized that using mix method approaches can be more complex to carry out as this require more expertise to collect and analyse and interpret data, than using one method would, and can require extra resources, but we have enough budget and a complete team of experts that have skills to conduct both type of studies.
We have added more information in the study design section to help the reader understand the reasons we used a mix-methods approach. We have also restructured the flow of the paper through the comments you and other reviewers made, which will hopefully make it easier to read and follow.  

Reviewer 4 Report

The article entitled “Effects of menstrual health and hygiene on school absenteeism and drop-out among adolescent girls in rural Gambia” is interesting and according to the scope of the journal. It is publishable in the journal after addressing a Major Revision.

  1. I suggest providing the year of data collection in the abstract.
  2. Structure of the article should be added at the end of the introduction.
  3. In Figure 1, you have to write the number of sample sizes in the parenthesis.
  4. I am not satisfied with the Conclusion. The structure of the conclusion has not been properly written. You must have to follow this structure. Firstly, write the main aim of the study. Secondly, you have to write about the statistical methods that have been used to approach study objectives. Thirdly, the main findings of the study. Lastly, write about policy implications.

Author Response

Dear Reviewer,

Thank you for your time reviewing the manuscript and providing valuable feedback.

  1. I suggest providing the year of data collection in the abstract.

Answer: This has been added

  1. Structure of the article should be added at the end of the introduction.

Answer:  We have restructured the introduction, we have ended with highlighting the main gap, why the study was done, and the objectives. The objectives are listed in the way that the results are structured.

3. In Figure 1, you have to write the number of sample sizes in the parenthesis.

Answer: Figure 1 has been amended.

4. I am not satisfied with the Conclusion. The structure of the conclusion has not been properly written. You must have to follow this structure. Firstly, write the main aim of the study. Secondly, you have to write about the statistical methods that have been used to approach study objectives. Thirdly, the main findings of the study. Lastly, write about policy implications.

Answer: We have added more information to the conclusion section, including a summary of the main findings and adding more information about policy implications.

Reviewer 5 Report

The topic is interesting and relevant because of the implications it has on the education of girls, and on their future as full-rights citizens, however, the methodology presented has gaps that should be resolved. The interview process is not detailed enough. The analyzed questionnaire has not been previously validated, or the process or results of said validation are not indicated.

The layout of the tables is confused, which makes it difficult to interpret. On the other hand, the description of the results is scarce and not very relevant, leaving the information in the aforementioned tables. The qualitative results are described based on the questions presented; a sufficient categorization is not made to justify the subsequent statistical analysis.

The discussion and conclusions are also not very relevant, being limited to a comparison with other studies, but without providing new information beyond the raw data found.

For all this, an in-depth review is recommended before publication.

Author Response

Dear Reviewer,

Thank you for your time reviewing the manuscript and providing valuable feedback.

Comment 1: The topic is interesting and relevant because of the implications it has on the education of girls, and on their future as full-rights citizens, however, the methodology presented has gaps that should be resolved. The interview process is not detailed enough. The analyzed questionnaire has not been previously validated, or the process or results of said validation are not indicated.

Answer: We have added more information on the interview process and validation process in section 2.3.1, section 2.4.1, section 2.4.5.  We hope now is more clear for readers.

Comment 2: The layout of the tables is confused, which makes it difficult to interpret. On the other hand, the description of the results is scarce and not very relevant, leaving the information in the aforementioned tables. The qualitative results are described based on the questions presented; a sufficient categorization is not made to justify the subsequent statistical analysis.

Answer: We have reformatted the tables and reworded some parts to hopefully make them easier to go through.  We have added more in text information in the results. The justification for the statistical analysis used is in section 2.5.2

Comment 3: The discussion and conclusions are also not very relevant, being limited to a comparison with other studies, but without providing new information beyond the raw data found.

Answer: We aimed to discuss all the main points that the results showed as being significantly linked to missing school while menstruating, and how these factors can be addressed or the implications of the factors. We have improved the conclusion to bring together the points in the discussion to show implications of the findings and next steps.

Round 2

Reviewer 1 Report

The Authors improved the paper according to the suggestions of my previous review report. Now the paper is suitable for the journal.

Congratulations!

Author Response

Thank you for reviewing our manuscript.

Reviewer 3 Report

After reviewing the author responses, the author successfully addressed most of the comments and suggestions. Before publication, I recommend authors to supplement "Development of the survey and interview form" for future researchers and add these form as a supplementary files.

Author Response

Thank you for taking the time to go over our manuscript again. We have added a bit more detail in the manuscript about development of tools in section 2.3. We will also add the tools as supplementary files. 

Reviewer 4 Report

Authors have addressed all previous comments, and now paper is ready to publish in the journal. 

Author Response

Thank you for reviewing our manuscript.

Reviewer 5 Report

The authors have improved the deficiencies so that the document would be suitable for publication. Despite the fact that the sample is prior to the pandemic, which devalues the results.

Author Response

Thank you for reviewing our manuscript.